# Nanotechnology-Based RNA Vaccines: Fundamentals, Advantages and Challenges

**DOI:** 10.3390/pharmaceutics15010194

**Published:** 2023-01-05

**Authors:** Vitaly P. Pozharov, Tamara Minko

**Affiliations:** Department of Pharmaceutics, Ernest Mario School of Pharmacy, The State University of New Jersey, Piscataway, NJ 08854, USA

**Keywords:** RNA, coronavirus, vaccine, nanotechnology, nanomedicine, immunity

## Abstract

Over the past decades, many drugs based on the use of nanotechnology and nucleic acids have been developed. However, until recently, most of them remained at the stage of pre-clinical development and testing and did not find their way to the clinic. In our opinion, the main reason for this situation lies in the enormous complexity of the development and industrial production of such formulations leading to their high cost. The development of nanotechnology-based drugs requires the participation of scientists from many and completely different specialties including Pharmaceutical Sciences, Medicine, Engineering, Drug Delivery, Chemistry, Molecular Biology, Physiology and so on. Nevertheless, emergence of coronavirus and new vaccines based on nanotechnology has shown the high efficiency of this approach. Effective development of vaccines based on the use of nucleic acids and nanomedicine requires an understanding of a wide range of principles including mechanisms of immune responses, nucleic acid functions, nanotechnology and vaccinations. In this regard, the purpose of the current review is to recall the basic principles of the work of the immune system, vaccination, nanotechnology and drug delivery in terms of the development and production of vaccines based on both nanotechnology and the use of nucleic acids.

## 1. Introduction

Despite the stubborn resistance of certain groups of the population and organizations, vaccination is now widely used to prevent and fight various infectious diseases [1,2]. A wide variety of different types of vaccines are being developed and tested today starting with classical live attenuated vaccines [3,4,5,6], viral vectors [7,8] and ending with more recently developed vaccines using nucleic acids [9,10,11,12,13,14,15,16,17,18,19]. Regarding the coronavirus only, more than 100 vaccine candidates against COVID-19 are currently undergoing development, and about 40% of them reached clinical trials [20,21]. Design of vaccines based on the use of nucleic acids delivered by nanoparticles currently attracts the most attention of researchers [22,23,24,25]. It is understandable that the development of this type of vaccine requires the involvement of researchers of very different specialties, including virologists, immunologists, specialists in drug delivery, nanotechnology, molecular biologists, and so on. The fruitful cooperation of scientists from such different specialties is often hampered by the fact that they speak different scientific specialized languages. In addition, in order to quickly achieve positive and important results, knowledge of at least the basic principles of all the listed scientific specializations is required for each participant of such a multidisciplinary group. mRNA vaccines represent much suitable technology for a rapid scale-up production when compared with traditional vaccine methods. However, scalability of existing already tested and working vaccine also represents a substantial challenge [26]. It not only requires hundreds of different inputs, but involves extensive technology transfer and process set-up technology transfer between various groups of workers with different educational background. We hope that this work will be interesting and at least to some extent useful for a wide range of readers with various scientific and technical backgrounds and also provide a useful didactic material for education.

The current review, after general basic introduction, includes essentials of immunity, nucleic acids and nanomedicine. In this section, we present a brief overview of: (1) innate and adaptive immune systems; (2) organs, tissues, cells and molecules involved in the development of immune response and accountable for killing of external invaders (as well as of some internal malfunctioning cells); (3) types and general mechanisms of immune response and formation of immune memory and immunity per se; (4) forms and nature of immunity, difference between naïve and immune individuals, as well as, population or herd immunity; and (5) types of vaccination and vaccines (Figure 1). This part of the review, in our opinion, is necessary for a more complete and in-depth understanding of mechanisms of action of various vaccines and immunity development which are in turn essential for the development of new effective vaccines in general and vaccines based on the nanotechnology approaches in particular.

The next part of the review is devoted to highlighting roles of nucleic acids in protein synthesis as well as the brief explanation of therapeutic actions of nucleic acids. Since DNA and RNA play a very important and fundamental role in the synthesis of proteins, they can be used to trigger the synthesis of proteins specific to the virus (antigens), which in turn can trigger an immune response and ultimately lead to the development of long term and stable immunity. Initiation of immune response by nucleic acids seems to be more rational and safer than the use of live attenuated viruses or their parts. However, the lifespan of nucleic acids under adverse body conditions and their ability to penetrate the plasma membrane of cells are usually insufficient to elicit a sustained and significant immune response before they are fragmented and eliminated under the influence of various enzymes, relatively high temperature, oxidation and other adverse conditions of the internal body environment. In this regard, it becomes necessary to use methods for protecting nucleic acids and delivering them into the cytoplasm of antigen presenting cells where they can initiate the expression of specific proteins on the surface of the cell membrane and thereby trigger the immune response of the body. The nanotechnology can provide such an opportunity. Delivery of nucleic acid-based vaccines using nanoparticles allows to successfully solve all of these tasks, namely increasing the stability of genetic material by protection of DNA or RNA from degradation and augmenting their ability to penetrate inside the cells and effectively transfect the host.

Therefore, the next part of our review is dedicated to the description of nanoparticles as specific types of colloidal systems, brief discussing of their characteristics, classification, as well as a brief representation of the advantages and disadvantages of nanotechnology-based delivery systems in terms of their use for vaccination.

Finally, the last part of our review, prior to the conclusions, is devoted to the actual discussion of nanotechnology-based nucleic acid vaccines, their most effective types, advantages and disadvantages. We will consider these topics mainly on the examples of such a burning problem as vaccines against COVID-19.

## 2. Essentials

### 2.1. Immune System

The main task of human immune system is to protect the body from external (exogenous) invaders and internal (endogenous) pathogens. The system is not localized in one organ or one single part of the body. In contrast, it consists of a wide network of various actors including organs, tissues, cells and molecules creating together an orchestra which individual parts synchronously work together to protect the body (Figure 2). The life cycle of immune cells starts in the bone marrow as hematopoietic (blood forming) stem cells (the root of all specialized blood cells), more specifically lymphoid stem cells that form lymphoblasts [27,28,29]. A part of these immature cells forms so-called B-lymphocytes (B-cells), the immune cells that produce antibodies. Natural Killer (NK) cells also originate from common progenitor cells, i.e., hematopoietic stem cells (HSCs) during the transition from CD56high into CD56low; they undergo a progressive loss of NKG2A and expression of KIRs, CD57, and NKG2C on terminally differentiated NK cells. Another part of the immature stem cells produced by the bone marrow enters the blood stream and become mature T-lymphocytes in the thymus, an organ located in the upper chest. These three types of cells together with granulocytes (basophils, neutrophils and eosinophils that are distinguished by their affinity to different types of cell death) form so-called white blood cells [30]. The tonsils (two round lumps in the back of the throat), adenoids (lumps of tissue up behind the nose on the so-called soft palate), Peyer’s patches (groupings of lymphoid follicles in the mucus membrane of small intestine) spleen and thymus together with lymphatic vessels are the primary parts of the lymphatic system [31]. The blood stream also contains a vast amount of white blood cells (usually between 4000 and 11,000 per microliter of blood) and also plays a substantial role in human immunity while is normally considered as a part of the cardiovascular system. It should also be stressed that the so-called reticuloendothelial system (RES) [32]. RES is a network of immune cells comprising circulatory phagocytes (e.g., monocytes, neutrophils, dendritic cells, etc.) and tissue-resident phagocytes (Kupfer cells in the liver, alveolar macrophages in the lung, microglia in the brain, histiocytes in the connective tissue, red- and white-pulp macrophages, marginal zone macrophages, and marginal zone metallophilic macrophages in the spleen, etc.). In addition to these immune cells, specialized endothelial cells (liver sinusoidal endothelial cells) play an an essential role in the clearance of foreign particles/materials, viruses, and endogenous soluble substances in the circulation and tissues [33,34,35]. RES therefore can also be considered as an important part of immune system. Moreover, the liver, which produces the most proteins in the so-called complement system (a heat-labile component of normal plasma that “complement” the antibacterial activity of antibody) [36] might also be considered as one of the immune organs.

#### 2.1.1. Innate and Adaptive Immune Systems

Although all players of the immune system tightly work together, two broad types of immune responses (as well as immune systems) are generally distinguished: the innate immune system/response and the adaptive immune system/response [29,37]. The innate, or non-specific, immune response is an immediate and generally non-specific reaction of immune system on any disease-causing (pathogenic) invader. Together with skin, mucous membranes as well as complement system and RES, the innate immune response represents a first line of defense against common pathogens (bacteria, viruses, germs, etc.). The main purpose of this immediate defense is to prevent the spread of foreign pathogens all over the body. It should be stressed again that the response of the innate immune system to the invasion is non-specific and therefore is similar for different pathogens. Moreover, this system does not form a memory about the invader and response. Consequently, for all different exogenous pathogens, the response is very similar [29].

In contrast to innate immune system, adaptive immune system (also referred to as acquired immunity or specific immunity) represents a second line of immune defense and is specific to the external stimuli [29]. It relies on the memory formed after a previous attack of a specific invader. Few memory cells are conserved after the attack of an invader was repulsed. These memory cells preserve the antigen receptors that quickly recognize previously flighted invaders. So called clonal expansion of these few preserved memory cells rapidly produces millions of T and B lymphocyte clones capable to fight a specific previously seen exogenous pathogen [38]. It is preserved memory cells that provide immunity to already faced exogenous disease-causing attackers and form a long-term immunity after survival infections. Consequently, vaccinations are aimed at forming an immune memory about a specific pathogen (without provoking an actual disease) in order to mobilize a strong immune response after following actual infection.

#### 2.1.2. Organs, Tissues, Cells and Molecules of the Immune System

The functioning of the immune system is supported by immune cells and protein molecules. Innate immune response is provided by innate lymphoid cells (ILC) and innate-like lymphocytes (ILL) (Figure 2). The collective term ILC is used for immune cells with lymphoid morphology which do not express T-cell receptors (TCR) and therefore are not capable to recognize antigens—foreign substances that induce an immune response (e.g., chemicals, bacteria, viruses, toxins, etc.) [39]. ILCs are responsive to cytokines (small signaling proteins/peptides) and generate other cytokines or interferons that activate other innate or adaptive lymphoid cells and can induce various changes in homeostasis, inflammation, tissue remodeling and stimulate immediate (and later adaptive) immune defense against different pathogens. ILLs express invariant T-cell receptors and together with non-TCR-expressed ILCs form a large group of natural killer cells controlling microbial infections and tumor growth. In addition, innate lymphocytes play a regulatory role in involved in reciprocal interactions with T cells, macrophages, dendritic and endothelial cells [40].

Adaptive (acquires or specific) immunity provides the second line of defense by the clonal expansion of T and B cells (Figure 2). Each clone of adaptive lymphocytes expresses the same antigen receptor as the original memory cells, and therefore all cloned cells fight the same pathogen [37]. Such clonal expansion of adaptive immune cells is a hallmark of adaptive immune response and can be found only in vertebrates including humans. B cells produce antibodies—Y shaped proteins (immunoglobulins) which mark invader cells for destruction by other immune cells through binding to the surface of exogenous pathogen cells (and some types of internal cancer cells). Each B cell produces antibodies that are highly specific to only one type (or very similar types) of pathogens. Regulatory B cells, similar to innate lymphocytes) can also release cytokines inducing various immune responses. T cells recognize antigens and either stimulate B cells to produce antibodies (helper T cells) or directly kill cells infected by an exogenous attacker.

In addition to cytokines and antibodies, the third major part of molecules related to the immune response are proteins of the complement system (Figure 2). Proteins from this system augment so called opsonization of pathogens by antibodies [41,42,43]. Opsonization is a process that uses opsonin (e.g., antibody in case of immune system) to prepare a molecule (e.g., antigen or invader) attractive to the phagocytosis. For instance, binding of antibodies (opsonins) to the surface of a bacteria attracts phagocytes (natural killer cells or macrophages) to take up and kill opsonized pathogen cell. In other words, some blood plasma proteins “complement” the antibody antibacterial activity defining the name of the entire system. Other proteins of complement system can send signals which attract neutrophils to sites of infection or form a complex with a microorganism destroying the latter. Therefore, proteins of complement form a bridge between the two arms of immune system.

In summary, all cells and molecules involved in both innate immediate not-specific and adaptive highly specific immune responses are tightly and cooperatively working together to fight infections and diseases.

#### 2.1.3. Adaptive Immune Response

As mentioned above, the adaptive immune response to an infection not only fights and destroy the intruder but also creates a memory about the invasion and mechanisms of effective defense against the exogenous pathogen. Such a memory is achieved by preserving a blueprint of ammunitions effective against the invader by saving a small part of lymphocytes specific to the pathogen. It is very important that immunological memory allows the immune system to respond rapidly and effectively to the previously encountered pathogen [29,37]. Consequently, in order to generate such a memory without severe disease (or with its mild symptoms), one can deceive the immune system and simulate an adaptive immune response by initiating or stimulating various critical steps in the formation of an adaptive immune response, which is generally achieved by the process called vaccination. Consequently, it is important, at least very briefly, to note the main stages of the adaptive immune response.

Activation of a specialized antigen-presenting cells (APC) is the first step in the initiation of adaptive immunity (Figure 3). Antigen is denoting an extraneous molecule that initiates immune response by reacting with immune cells [44,45]. Antigen must be a foreign alien (“non-self”) molecule. In the course of genotypic (evolution) and phenotypic (during the life span of an organism) certain types of harmless foreign antigens (e.g., pollen, dust, food proteins, etc.) become acquainted “self” molecules that do not trigger immune response (in the absence of pathological immune reactions like allergy). The process of the suppression of immune reaction to known relatively damaging molecules is called tolerance [37]. The immune system contains specialized cells that detect potentially harmful molecules. These cells engulf the invaders and present antigens and some specific immune signaling molecules on their plasma membrane. For instance, the bacterium absorbed by a macrophage (specialized cells that detect, phagocytized and destruct bacteria and other dangerous organisms) is encapsulated in a vacuole, digested by lysosomes and releases antigen(s). The antigen is moved (presented) to the surface of the macrophage coupled with the so-called major histocompatibility complex (MHC) class I and II molecules [46]. Consequently, the cells that show (present) antigens on their plasma membrane are called as (APCs. Macrophages, B cells in an early stage (before activation and differentiation) and especially dendritic cells represent the major types of APCs. These cells are presented in the skin, nose, lungs, stomach and intestines which are often in contact with exogenous molecules. Ultimately, these tissue-resident APCs migrate through the lymph to a local lymph node where the presented molecules are recognized by naïve T of B cells (Figure 3, left panel). In addition to MHC I and II, several other transmembrane proteins can be expressed in the plasma membrane of APCs (some of these molecules are shown as examples in Figure 3, left panel). In addition to plasma membrane proteins, several types of interferons and other signaling cytokines are also excreted. All these signals eventually interact with recirculating naïve T and B lymphocytes. These lymphocytes activate and produce clones of activated antigen-specific T and B cells which then migrate to the site of infection (Figure 3, right panel). The activated T cells induce apoptosis (programmed cell death) after the recognition of the antigen in infected cells and APCs which finally are safely removed from the body. B cells generate antibodies that also induce apoptosis in infected cells. In addition, interferons, cytokines and chemokines also take part in killing of infected cells.

Theoretically, adaptive immune response and immune memory can be triggered on several stages of formation of adaptive immune response. Weakened low potent infectious cells, their parts or even proteins can be introduced into APCs on the skin, nose, lungs or even orally. This may trigger the adaptive immune cascade from the very beginning. Antigen can be somehow delivered into APCs leading to the expression of MHC and activation of lymphocytes. Antigen can be substituted with DNA or mRNA expression vector encoding this protein giving a similar result. Corresponding MHC or other epitopes can be forced to be presented by APCs by transfecting APCs with DNA or mRNA encoding these proteins. Ligands for TCRs or other signaling receptors may be delivered to naïve helper T cells to activate them. Cytokines, interferons or/and chemokines responsible for the triggering of adaptive immune response can be used. Finally, entire activated T or B cells can be injected. These are just few theoretically possible approaches for the initiating of the immune response and immune memory to the specific virus or other microorganism. Later in the manuscript, some of these methods will be discussed with examples.

### 2.2. Immunity

In simple plain terms immunity to infection can be defined as the ability of an organism to resist or fight infection. Several types of immunity are generally distinguished including innate, natural, adaptive, active, acquired, passive, etc. [47,48]. The strict borderlines between these types of immunity are still not clearly defined and different authors often use different terms interchangeably. In our opinion, in the scope of the present manuscript, it is important to distinguish innate or intrinsic immunity that most of us have since the birth and acquired or adaptive immunity which is developed though the life of an individual. The latter can be passively borrowed from another source (e.g., transient short-term immunity obtained by a child from the mother with breast milk or via antibody-containing blood products) or actively acquired as a result of a successfully survived disease or by vaccination. Here, we are especially interested in the last type of adaptive immunity.

#### Naïve versus Immune Individuals

The individual which was not previously infected and survived a certain contagion usually does not have intrinsic or adaptive immunity against this particular invader. Consequently, if a virus (or some other infectant) enters the body and survives innate non-specific immune defense and reaches the targeted organs or cells, it begins to reproduce itself in many copies. During this process of reproduction, several toxic substances usually are generated and poison the host cells, organs and being secreted in the blood stream reach and affect other organs and cells. In addition, the host body cells often stop performing their normal functions and begin to help the invaders to reproduce themselves. The negative results of such processes typically appear as first symptoms of the disease and normally lead to the activation of adaptive immune response. Combined with negative poisoning effects of the process of reproduction and vital activity of contagion agents, the activation of innate and adaptive immune system usually induces disease symptoms (many times severe) that make infected person suffer from a fever, malaise, headache, rash, etc. Moreover, often, an inadequate and redundant secretion of cytokines, recruitment of inflammatory cells and products of pathogen killing lead to severe damaging of body homeostasis and function which can even cause the death of the infected person [49]. One of the typical examples of such adverse reaction includes the development of so-called cytokine storm or the cytokine release syndrome—a life threatening condition of elevated levels of circulating cytokines and immune-cell hyperactivation that can be caused by various therapeutic interventions, autoimmune disorders, cancers, etc. [50]. Cytokine storm attracted special attention during COVID-19 epidemics when this overreaction of immune system caused serious health problems [10,51]. In summary, infection of a “naïve” to a particular invader individual in most cases causes a wide spread of pathogen in the body, serious health problems and possible long term adverse health consequences.

In contrast, immune person (vaccinated or previously survived the same infection disease) already has prepared immune cells targeted for a specific exogenous pathogen. In case of recurrent contact with known pathogen, these prepared immune armors are rapidly multiplied and generated an adequate immune response that in most cases minimizes spreading of the invader throughout the body and limits adverse effects of the invasion. This protective effect of developed adaptive immunity serves as a main justification of the vaccination. The second objective reason for vaccination is the development of cooperative (so-called “herd”) immunity in an entire population that limits spreading of the contagious disease and finally leads to termination of pandemic.

During an epidemic, each particular individuum, as well as the entire society, is extremely concerned in preventing or at least limiting the spread of contagion. From the point of view of the ordinary person, it is important not to get infected yourself or at least survive the infection with minimal adverse consequences. From the point of view of the entire community, it is imperative to stop a wide uncontrolled passing the infection between the members of the group. Such task can be achieved mainly by two approaches: social distancing and herd immunity [49]. Calculations have shown that in the case of a 100% effective vaccine, a homogenous population and random vaccine coverage, the percent of vaccinated among the entire population must reach around 70% to achieve herd immunity and reduce the cumulative lifetime incidence of infection in unvaccinated individuals to zero [52]. Unfortunately, no community is completely homogeneous, just as no vaccine is 100% effective. As a result of these imperfection, a much larger percentage of vaccinated is required for the development of herd immunity and this percentage varies from community to community [52,53].

### 2.3. Vaccine Types

The main objective of vaccination is to acquire a long-term adaptive immunity while preventing the development of a serious illness caused by the introduction of pathogens into the body of a recipient. For these purposes, several types of vaccines are used [54]. Live vaccines usually contain alive attenuated (weakened or inactivated) replicating strains of the corresponding pathogens (Figure 4). Such weakened pathogens should replicate sufficiently to elicit an immune response, but not enough to induce significant disease manifestations. Similar result can also be obtained with an appropriate number of killed whole organisms of pathogens. However, introducing an entire weakened or even dead pathological organism into the body of the recipient still potentially can induce the disease. In order to prevent such adverse reactions, only certain parts of the pathogen which are able to induce adaptive immune response (e.g., purified structured or recombinant proteins, polysaccharides, toxoids, inactivated protein toxins, etc.) can be utilized (Figure 4). Unfortunately, such small fragments typically demonstrate poor pharmacokinetics and stability in the aggressive environment of the body. Therefore, in order to protect them from the degradation and improve pharmacokinetics of vaccine, different nanotechnology-based approaches, discussed later, can be applied.

### 2.4. Nucleic Acids

#### 2.4.1. DNA and RNA

In order to better understand the mechanisms of action of nucleoside-based vaccines, it seems necessary to at least briefly dwell on the types of nucleic acids and their role in the synthesis of proteins, which in turn form the basis of the vital activity of an entire life on our planet. In general, nucleic acids are naturally occurring macromolecules that are actually biopolymers composed with phosphoric acid, 5-carbon sugars and organic bases (purines and pyrimidines) as shown in Figure 5. Two main and the most important for the topic of the present review classes of nucleic acids are deoxyribonucleic acid (DNA) and ribonucleic acid (RNA). Each nucleotide consists of a nitrogen-containing aromatic base attached to a five-carbon sugar (pentose), which is in turn attached to a phosphate group. Such sugars are bonded together via phosphate groups arranging in the backbone which in turn forms a helical chain (Figure 5).

The internal structures of DNA and RNA are very similar in many ways [55]. Both of these biopolymers contain long chains of sugars linked together by phosphate groups (Figure 5). However, there are two significant differences. First, DNA contains two such chains interconnected through complimentary nucleotides via hydrogen bonds forming one double helical structure. The second vitally significant difference between RNA and DNA is that pentose in DNA is represented by deoxyribose (hence its name is deoxyribonucleic acid), while in RNA (ribonucleic acid) this sugar is represented, respectively by ribose [55]. Due to the fact that ribose has an additional oxygen atom that forms an additional hydroxyl group, RNA is much less stable than DNA. Since RNA mainly works in the cell as a short living messenger that transmits the information contained in the genes to other players in the protein synthesis, there is no need for great stability of the RNA. However, when RNA is employed to trigger the immune response (just like in other therapeutic applications), this instability represents a great challenge. Consequently, in order to defend the instable RNA against the adverse effects of the biological environment inside the body specific delivery vehicles that protect the cargo, for instance, nanocarriers, can be used [10,56,57,58,59].

The process of protein synthesis involves two main phases (Figure 6). In the first phase called transcription, the information encoded in DNA is copied into the genetic code of the newly formed messenger RNA molecule [60].

#### 2.4.2. Therapeutic Action of RNA Constructs

The therapeutic effect of RNA constructs depends on their influence on protein synthesis. Exogenous RNA, when delivered to the cell cytoplasm (e.g., by nanoparticles), can both suppress and stimulate protein synthesis. For instance, a double-stranded small interfering RNA (siRNA) or single-stranded micro-RNA (miRNA) do not themselves code for any proteins. Nevertheless, they can inhibit the synthesis of a particular protein through the mechanisms of the so-called RNA interference (Figure 7A). Such suppression of protein synthesis occurs via the RNA interference pathway, where the newly synthesized in the nucleus mRNA is degraded in the so-called RNA-induced silencing complex (RISC) preventing further translation of the information embedded in this daughter mRNA into the protein. On the contrary, delivery of a single-stranded coding RNA within the cell is capable of initiating protein synthesis by translating the information embedded in delivered mRNA into a protein with the participation of the intracellular machinery of protein synthesis [61]. Depending on the information that is embedded in the delivered mRNA, the synthesized protein can either perform its functions inside the cell or be secreted out from the host cell (Figure 7B). It can also be expressed on the cell membrane (e.g., as a receptor) or present major histocompatibility complex (MHC) or antigen on the cell surface [62]. It is in the latter case that mRNA can be used for the initiation of the immune response and, ultimately, the development of long-term immune memory performing thus functions of RNA vaccine. It should be stressed again that only coding RNA sequences can be utilized as vaccines, while other types of RNA constructs cannot.

### 2.5. Nanomedicine

Although nanomaterials have been used empirically centuries ago by crafters, their first scientific description was made by Michael Faraday in his classical the Bakerian Lecture for the Royal Society of London published in 1857 [63] where he reported the formation of fine nanosized gold particles. In 20th century, advances in the development of robust methods of fabrication and accurate methods of characterization and visualization of nanoscale sized objects led to the burst development of novel branches of science and technology, such as microelectronics, biochemistry and molecular biology. Inevitably, the success of nanotechnology resulted in the attempts of using its approaches to the diagnostics, treatment and prevention of diseases. Finally, nanomedicine as a branch of medicine have emerged. The first use of the word “nanomedicine” in scientific literature usually dated to the last decade of the previous 20th century [64,65]. Therefore, nanomedicine is relatively young but reached probably the most productive age where a novel scientific and technology area (as well as human being) reaches a peak of scientific productivity. The success in application of nanotechnology approaches to medicine may best be evaluated by the number of publications dedicated to nanomedicine. A simple PubMed search returns more than 3000 of publications with the word “nanomedicine” in the manuscript title and more that 100,000 papers where this word is used elsewhere in the text. According to the Oxford Dictionary, nanomedicine represents a branch of medicine that applies the principles and approaches of nanotechnology to diagnose, treat and prevent diseases [66].

#### 2.5.1. Nanoparticles

The first part “Nano” in the word “Nanomedicine” usually denotes either the application of nanotechnology for medical purposes or a dosage form prepared using nanoparticles. In relation to the topic of our review, the second definition is more suitable for describing RNA and DNA vaccines constructed with the help of nanoparticles. In terms of SI units of measure, the term “nano” has the specific sense “one billion” (10^−9^) [67]. For the particle size, one nanometer is equal 10^−9^ m. Nanoparticles are usually defined as particles with overall volume that is larger than atomic dimensions but are small enough to exhibit Brownian motion [68]. In this sense, they may be considered as colloidal particles because nanoparticles resemble all characteristics of colloids (Figure 8). Figure 8 shows the absolute and relative sizes of various nanoparticles.

Colloids are usually defined as materials which are homogeneously dispersed in another substance(s) [69]. In other words, colloidal particles must be larger that atomic dimensions but should be small enough to demonstrate Brownian motion. It is generally assumed that colloidal particles have a size from 1 to 1000 nm (10^−9^–10^−6^ m) at least in one direction. Both, the dispersed materials (the component of colloid found in the lesser extent) and dispersion medium (the constituent of colloidal system found in the greater extent) can be in the solid, liquid or gaseous state forming eight (excluding the gas-gas type system which forms a completely homogeneous at the molecular level solution) different types of colloids which do not settle and cannot be separated by ordinary filtering and centrifugation. More strict classifications of colloids define colloidal particles as non-crystalline materials [70]. However, recent advantages in liquid crystals led to emerging of so-called “liquid crystal colloids” that represent a fast-growing research arena in a crossing point between traditional colloids and liquid crystals. Nanoparticles which are being used or investigated for medical use, also resemble a type of colloids and by consensus are defined as particles with size between 1 and 100 nm [71]. However, based on recent advances in nanomedicine (including nanoparticle-based vaccines), in our opinion, particles with sizes between 1 nm and 1000 nm (1 µm or 10^−6^ m) should be considered as nanoparticles because they possess all features of colloids.

In most cases, unloaded nanoparticles in their native form cannot be considered as drugs, because therapeutical nanoparticles usually are designed to be inert and should not cause biological effects until a free drug or other active ingredient is released from the entire system and penetrate a cell or interact with a cell in some way [72]. However, it should be noted that sometimes nanoparticles are developed as toxic components, for example, used for example in the treatment (chemotherapy) of cancer (so-called “drug-free macromolecular therapeutics”) [73].

There are many different types of classifications of both colloids and nanoparticles. They are classified by size, type and aggregate state of the materials from which they are made. Additionally, nanoparticles can be characterized by electric charge and magnetic properties, spatial structure, architecture, and many other factors. From the point of view of the delivery of nucleic acids, including vaccinations, lipid-based and polymeric nanoparticles have the greatest prospects [10,22,24,56,57,74,75].

Among the lipid nanoparticles used for the delivery of nucleic acids, liposomes and the nanostructured lipid carriers (NLC) are most often used for this purpose [58,76,77,78]. Liposomes are nanovesicles consisting of a lipid bilayer (less often several bilayers) to a large extent to the plasma membrane of a cell and the fluid (most often saline or buffer) inside this membrane (Figure 9). The delivered substances can be included both inside the liposome in the aqueous phase (if it is water soluble), in the lipid double membrane (if it is lipid soluble and hydrophobic) or in both compartments of liposomes. NLCs are composed of both solid and liquid lipids as a core matrix [74,79]. NLCs are the most convenient vehicles for delivery of water insoluble drugs. However, NLCs (as well as liposomes) can also be used to transport active components of a different nature when the delivered substance located outside the nanoparticles is conjugated to the membrane of these vehicles by the specific binders, for instance, disulfide bonds [58,75]. Polymeric nanoparticles can be made using different natural or synthetic polymers. For instance, poly(lactic-*co*-glycolic) acid, a natural polymer chitosan, amphiphilic block co-polymers (having, like natural phospholipids, hydrophilic and hydrophobic parts) as well as other polymers can be used for preparing vesicles similar to liposomes and NLCs, polydispersed particles and nanoparticles with a highly developed structure such as dendrimers (Figure 9), as well as inorganic, metal, mesoporous silica nanoparticles, exosomes, etc. [80,81,82,83,84,85,86].

In terms of overall nanoparticle electric charge or surface charge (so-called Zeta potential), neutral or slightly negatively changed (anionic) vehicles are used due to the fact that positively charged (cationic) liposomes, as well as other cationic nanoparticles, exhibit significant cyto- and genotoxicity to human cells [72,76]. However, the delivery of nucleic acids using weakly negatively charged or neutral particles, although possible, is difficult. It is known that under physiological conditions, nucleic acids have a negative charge [24]. Therefore, it is natural to use a positively charged particle for their delivery. At the same time, it is possible to choose the ratio between negatively charged nucleic acids and cationic vehicles that they would be connected together tightly enough by electrostatic interaction and the resulting complex as a whole would be a neutral nontoxic nanoparticle [72]. In this case, a positive charge can be distributed both on the surface or/and inside the particle (Figure 9). A more complex design of the carrier particle is also possible and will be discussed below. It should be mentioned that the electrostatic interaction between a negatively charged nucleic acid and a positively charged particle is quite strong. On the one hand, the formation of a firm complex is helpful because it protects the resulting composite from decay, while moving inside the body, thereby protecting relatively unstable nucleic acids from degradation. However, on the other hand, such a strong interaction presents certain difficulties for the separation of nucleic acids from particles inside the cell, preferably into the cell cytoplasm so that nucleic acids can play its targeted role in cellular processes. In most cases, the internalization of relatively large nanoparticles by the cell occurs by endocytosis (Figure 10). Having entered the cell by endocytosis, the nanoparticle finds itself inside the endosome, which represent a membrane vesicle budded from the cellular plasma membrane. There are four main mechanisms by which RNA bound to nanoparticles can escape the endosome into the cell cytoplasm. In order to facilitate one or several of such mechanisms, different modification of nanoparticles and/or an entire complex have been proposed [87]. Without going into details, we only note that the use of these intracellular mechanisms, the release of RNA from its complex with a nanoparticle, makes it possible to prevent nucleic acids from being destroyed and allows RNA release into the cellular cytoplasm, where its intracellular action occurs.

#### 2.5.2. Advantages and Disadvantages of Nanoparticles for Medical Applications

The main advantages of the transport of nucleic acids in association with nanoparticles are: (1) the prevention of their destruction in the blood and intracellular fluids during their journey inside the blood or interstitial fluid and (2) the efficient delivery of nucleic acids into the intracellular cytoplasm intact and active. Previously, we showed that naked non-conjugated RNA degrades in human serum within minutes [58,92]. In contrast, conjugated to nanoparticles RNA remained largely intact after incubation for at least 2 days. The importance of delivery systems/nanoparticles was revealed for both major types of nucleic acids used in vaccines. It was shown that loading the mRNA into a polyplex micelle protected the mRNA from enzymatic degradation by 10,000-fold compared to naked mRNA [93,94]. In addition, it was found that the naked plasmid DNA was rapidly digested in the serum [95], whereas complexing it within the polyplex micelle substantially protected nucleic acid from enzymatic degradation [96]. In addition, the nucleic acid-nanoparticle complex can be targeted specifically to well-defined cell types, for example, in cancer treatment [58,72,92,97]. In addition, different types of nucleic acids of approximately the same size do not differ significantly from each other with regard to their complexing with the same type of nanoparticles. In terms of vaccine production, the latter means that the same or slightly modified manufacturing process can be used to produce a similar nanoparticle-based delivery system for different nucleic acids. Such a reuse of production technology greatly reduces the cost of production of novel vaccines.

The main disadvantage of medicines based on nanotechnology lies in the relatively high costs for the development and production of such formulations. In addition, medications based on the use of lipid nanoparticles often require special storage conditions to prevent their disintegration, such as low temperature or lyophilization. All of these factors kept nanomedicine from becoming widespread for a long time as pharmaceutical companies leaned towards producing cheaper (but less potent) formulations, such as lower molecular weight drugs. However, with the advent of the RNA-based vaccines against COVID-19, the advantage of nanotechnology in the production of effective vaccines has been convincingly shown [10,98]. As the old proverb says, “there would be no happiness, but misfortune helped”.

## 3. Nanotechnology-Based Nucleic Acid Vaccines

### Types of Nucleic Acid Vaccines

The main task of vaccines based on the use of the genetic material of viruses is to initiate an immune response without introducing the whole virus or its main proteins into the body. For this purpose, DNA or RNA encoding one of essential virus protein is delivered inside the cell to initiate the synthesis of the corresponding protein and an immune response. When developing DNA vaccines, it is important to take into account that deoxyribonucleic acid molecules in most cases must be delivered inside the cell nucleus where they can start the replication process leading to the formation of virus RNA, this RNA getting into the cytoplasm from the nucleus will start the synthesis of the corresponding viral protein which should then elicit an immune response from the body. In some rare cases, viral DNA can be transcribed outside the nucleus in extrachromosomal vesicles [99]. On the contrary, viral RNA must be delivered only to the cytoplasm to trigger the same effect and ultimately conduct to immune reaction. On the other hand, RNA, due to its features mentioned above, is less stable when compared with DNA. It should be however noted, that both DNA and RNA in principle require special delivery systems based either on viral vectors or nanoparticles for their effective delivery into the cell and, even more so, inside the nucleus [75,100,101]. In this sense, RNA, as a less stable molecule, requires more protection. In the following section, we will discuss vaccines based on the use of nucleic acids on the example of the COVID-19 vaccines.

## 4. COVID-19 Nucleic Acid-Based Vaccines

### 4.1. Structure of SARS-CoV-2 Virus

A fundamental task in the development of an acid-based vaccine is the choice of a protein specific to a given virus and capable of inducing a significant immune response of the body. Let us consider the selection of such protein on the example of coronavirus. Most coronaviruses are very similar in the structure and consist of single-stranded positive sense RNA enclosed inside a membrane envelope (Figure 11) [10]. Ironically, this structure resembles the RNA containing nanoparticle that is used to vaccinate against the coronavirus. Several proteins are used in the formation of the structure of the coronavirus, among which the most interesting is the spike protein that is responsible for the recognition and internalization of the virus by angiotensin-converting enzyme-2 (ACE-2) receptor expressing cells. It is not surprising that this particular protein was chosen as the antigenic protein and that it is the target of many other proposed approaches to treat or prevent coronavirus infection [10].

As mentioned above, the main task of using nucleic acids as vaccines is to initiate the body’s immune response without introducing the live or attenuated virus itself or its part. Therefore, the sequence of a delivered nucleic acid (DNA or RNA) is chosen to instruct the cell to synthesize the selected protein. In case of coronavirus this selected protein is most often spike protein [18].

### 4.2. DNA-Based COVID-19 Vaccines

Active research on DNA-based vaccines began about 30 years ago after the discovery of the experimental fact that plasmid DNA can induce the formation of antibodies against an encoded protein [102]. Typically, in order to prepare a DNA-based anti-coronavirus vaccine, a double stranded DNA is synthesized and cloned into a plasmid (Figure 12). The plasmid is usually delivered by intramuscular injection with help of an electroporation device to facilitate its uptake. Transfection of myocytes, tissue-resident macrophages or recruited antigen-presenting cells (dendritic cells) by the plasmid stimulates the synthesis of spike protein [103,104]. The expressed protein activates T cells through MHC class I or B cells directly. In turn, the necrosis-dead myocytes release spike proteins that activates CD4+ T cells through MHC class II presentation. Overall, all these processes trigger the activation of the immune system and eventually lead to the development of adaptive immune response and immune memory. The main advantage of DNA-based vaccines is the prevention of possible viral infection, as in the case of using the whole virus or part of it. In addition, such vaccines are significantly cheaper compared to protein-based vaccines and have increased stability during transportation and storage. Moreover, they can be administered to immunocompromised patients [105]. Unfortunately, DNA vaccines exhibit relatively low immunogenicity and usually requires multiple booster doses. In addition to this, theoretically, DNA could become a recipient’s genome and be inherited; the latter causes significant concern for patients leading to refusal of vaccination.

### 4.3. RNA-Based COVID-19 Vaccines

Unlike vaccines based on the use of DNA, RNA vaccines work in the cytoplasm, do not penetrate the cell nucleus, cannot integrate into the genome of the host and quickly degrade. A typical composition of RNA-based vaccine using lipid nanoparticles as delivery vehicles is presented in Figure 13. Lipid nanoparticles (LNPs) for these purposes are prepared through rapid mixing, often facilitated by microfluidic devices [107]. LNPs are formed through a cascade of merging smaller lipid vesicles containing mRNA encoding COVID-19 spike protein. Cationic lipids are used for creating of positively charged nanovesicles which form ionic interactions with the negatively charged mRNA. The resulting LNP-RNA complexes are typically referred as lipoplexes [108,109]. Such lipoplexes are usually covered with poly(ethylene glycol) (PEG) coats forming so-called STEALTH nanoparticles which are hardly recognized by the reticuloendothelial system, weakly interacts with blood and interstitial fluid components and therefore have an extended lifespan in the body [110,111,112,113]. In addition, STEALTH nanoparticles limit cyto- and genotoxicity of the payload [72,114,115]. The structure of mRNA used in the vaccines is similar to that in natural cellular RNAs. Both non-replicating mRNA and self-amplifying RNA are used in RNA-based vaccines [107]. Cellular non-replicating mRNA contains a so-called 5′ cap including the methylguanosine G moiety at the 5′ end, followed by a triphosphate linkage to the first nucleotide (Figure 14). Such structure plays several roles in mRNA translation and, in particular, protects the nucleic acid from the enzymatic cleavage inside cells. In addition to the use of a lipid nanoparticles as a delivery system, additional protection of delivered mRNA encoding SARS-COVID-19 spike protein, mRNA can be modified by replacing uridine to N1-methyl-pseudouridine. It was found that such modification in Pfizer-BioNTech and Moderna Therapeutics COVID-19 vaccines led to almost 2-times higher efficacy when compared with the vaccine used similar delivery system and similar but unmodified mRNA [116]. 5′ and 3′ untranslated regions (UTR) of mRNA located upstream and downstream of the coding region, respectively, are not translated into protein but also are involved in regulating expression of messenger RNA. At its 3′ end, mRNA contains a polyadenylated region which is called as poly(A) tail (Figure 13) which regulates the lifespan of this nucleic acid in the cytosol. Self-amplifying RNA also contains open reading frames encoding the components of RNA-dependent RNA polymerases (RDRP). RDRPs are used by RNA viruses for amplification of RNA [117]. Utilizing such a pathway in RNA-based vaccines multiplies the number of RNA copies in the host cell increasing the production of antigens. Adaptive immunity of vaccinated individual is activated by transfecting somatic and tissue-resident immune cells in the place of injection as well as immune cells in the secondary lymphoid tissues. Mechanisms of developing adaptive immunity by RNA-based vaccines have been discussed above.

### 4.4. Advantages and Disadvantages of RNA-Based Vaccines

RNA-based vaccines are flexible with respect to production and application (Figure 14. As was already mentioned, generally, the physico-chemical characteristics of RNA with approximately the same size are similar for different encoding proteins. Consequently, almost the same production process can be used for the manufacturing of RNA-based vaccines targeting different proteins. The latter substantially decreases the production cost of novel RNA vaccines based on the same delivery technologies [118]. RNA-based vaccines effectively induce the full complement of immune responses, including both humoral and cellular answers without injecting an entire virus or its parts into a recipient’s body. This approach prevents the possibility of developing an infection and other adverse reactions on the foreign cells, proteins and other parts of viruses. Moreover, the change of the targeted protein is achieved simply by changing the sequence of RNA making the formulation highly antigen adjustable with multipurpose applications. Among very rare adverse events of nanoparticle-based mRNA vaccines, anaphylaxis and other manifestations of hypersensitivity should be mentioned [119].

The major challenge (and in some extent—the advantage) for any vaccination technique is its high specificity for a contagious agent. On the one hand, a vaccine in order to induce a stable and specific adaptive immunity should be very specific to a particular pathogen. It is the specificity of a vaccine that mainly determines its efficacy when decrease in the specificity leads to the reduction in vaccine efficacy [120]. On the other hand, a high specificity of the vaccine can make vaccination less effective or even not effective at all against future invasions if the contagious agent changes its properties that are main targets for the particular vaccine (Figure 14). Virus mutations represent a perfect example of this situation. As was mentioned above, the sequence of RNA vaccine should be selected based on the composition of a major virus antigen. In case of coronaviruses, spike protein is an obvious choice because it represents the main characteristics of all viruses with the corona including COVID-19. However, viruses mutated frequently and often such mutations result in emerging a new persistent and viable variant of the virus [121,122,123]. It is possible that a mutation can change the structure of an RNA and protein targeted by a vaccine allowing the virus to acquire a partial or full immune escape. The omicron variant of coronavirus is an example of such a mutated virus [124]. However, the ability of rapid fine tuning of an RNA vaccine to a mutation in virus RNA by changing a structure of its RNA open a door for relatively fast development of more effective vaccine for such mutated viruses. For instance, Moderna has modified its mRNA-based COVID-19 vaccine to match the mutated sequence of the spike protein [116]. Moreover, nanoparticle-based delivery22z system allows for a delivery of several mRNAs (which in fact are already used in the modern effective vaccines) with different sequences to address existing mutations in the targeted viral protein. Other challenges in the development of RNA-based vaccines include a relatively high instability of vaccine formulations and possible multiple administrations. Low temperatures (−20 to −80 °C) are often required for long-term (3–6 months) storage of such vaccines which usually have a relatively short (~30 days) at 2 to 8 °C and a very short (2–24 h) shelf life at room temperature [125]. Another important factor that substantially influences antigenicity of a protein synthesized as a result of vaccination is the stability of this protein after synthesis, including conformational or fold stability in terms of maintaining its three dimensional and biologically active form [126]. Normally, such the stability and ability to initiate immune response of viral antigen is achieved by the assistance of the molecular chaperones [127,128]. Consequently, adding to a mRNA vaccine sequence(s) of other proteins that stabilize the main antigen and prevent its folding, can enhance the efficacy of the vaccine. In particular, it was already shown that some molecular chaperones including heat shock proteins possess extracellular immunostimulatory properties when complexed with antigens [129].

## 5. Future Directions

The main efforts in improving RNA-based vaccines include increasing the stability (in particular thermostability); preventing quite rare allergic reactions and possible adverted side effects associated with the use of poly(ethylene glycol) and impurity of lipid nanoparticles; preventing low effectiveness against new variants of the virus; significantly boosting the effectiveness of RNA vaccines by targeting nanoparticles specifically to antigen presenting cells; and finally improved vaccine administration. Several variants of COVID-19 RNA-based vaccines with relatively high temperature resistance and stability are being developed and tested in recent years [20,130]. It is also proposed to replace polyethylene glycol with other biopolymers such as *N*-(2-hydroxypropyl)methacrylamide (HPMA) copolymer and use ligands specific to dendritic cells (e. g, mannose or hydrophobic-interaction-inducing lipids) conjugated to the surface of nanoparticles [98,131]. It is well known that directing of nanoparticles specifically to their place of action significantly increases the accumulation of the delivered agent in targeted organs, tissues and cells, improves their pharmacogenetics and prevents unwanted adverse side effects [10,56,57,75,132]. One of the significant issues of lipid nanoparticle-loaded mRNA vaccines is off-targeting to the liver even though the LNP-mRNA vaccine is injected via intramuscular injection [133,134]. Such liver off-targeting may cause toxicity/immunogenicity and inflammation exacerbation based on the type of antigen that is expressed [135]. Minimizing hepatic off-targeting while maximizing the delivery at the injection site might be advantageous for widespread vaccine applications [136]. Efforts could also be directed towards improving the effectiveness of intramuscular vaccine administration and considering other routes of administration, e.g., oral or inhalation routes [98]. In particular, the delivery of various nanoparticles by inhalation already showed substantial advantages of this approach [36,58,77,92,100,137].

## Figures and Tables

**Figure 1 pharmaceutics-15-00194-f001:**
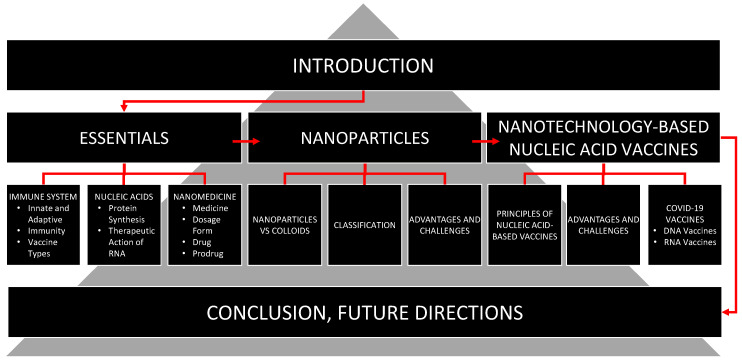
Structure of the review.

**Figure 2 pharmaceutics-15-00194-f002:**
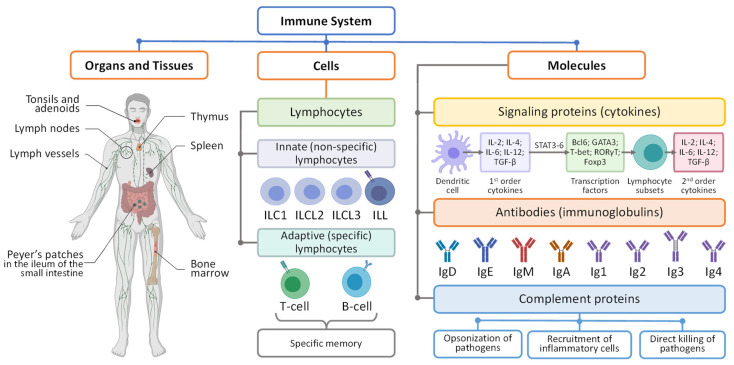
Overview of immune system.

**Figure 3 pharmaceutics-15-00194-f003:**
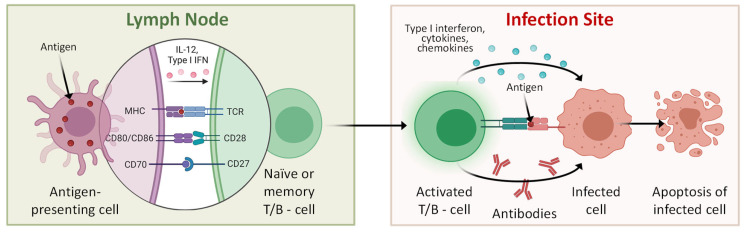
A simplified scheme of adaptive immune response.

**Figure 4 pharmaceutics-15-00194-f004:**
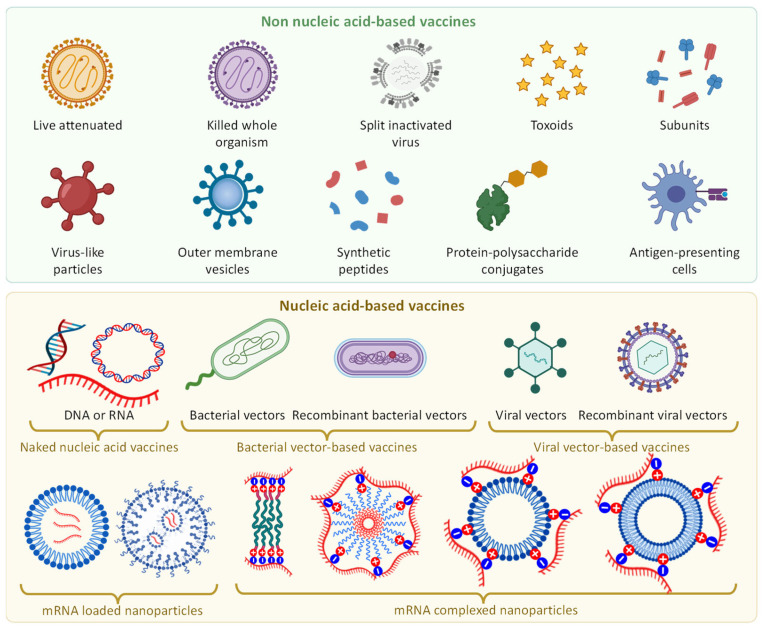
Different types of vaccine.

**Figure 5 pharmaceutics-15-00194-f005:**
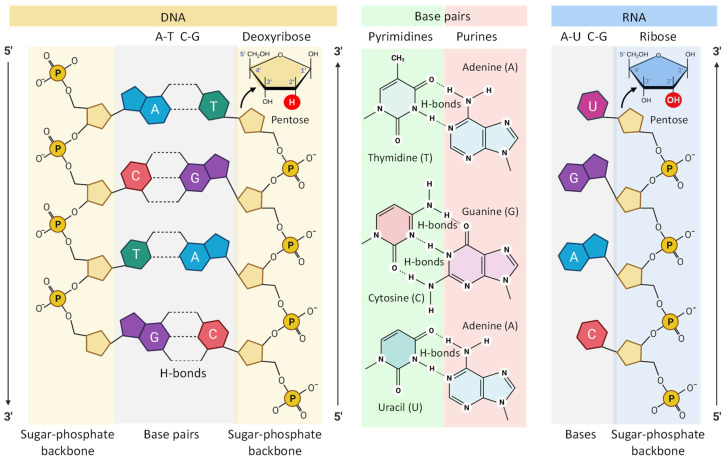
Molecular structure of DNA and RNA.

**Figure 6 pharmaceutics-15-00194-f006:**
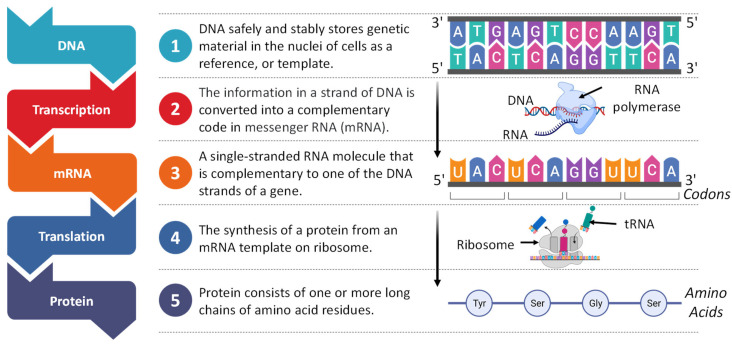
Main stages of protein synthesis.

**Figure 7 pharmaceutics-15-00194-f007:**
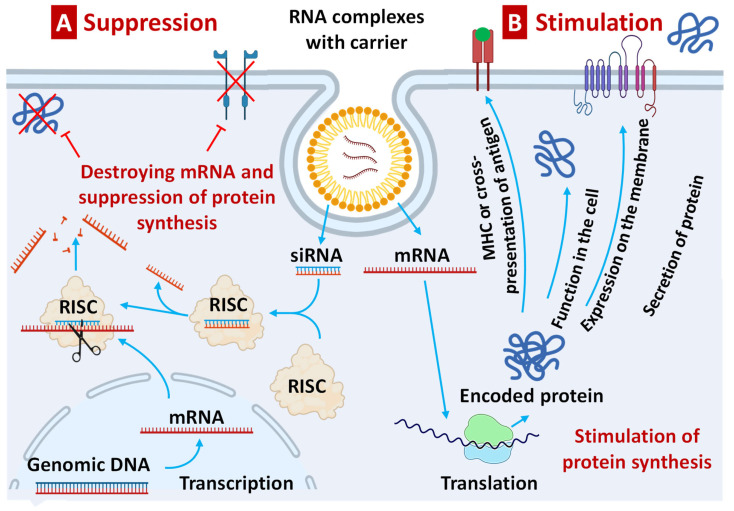
Therapeutic action of RNA constructs.(A) Suppression of protein synthesis by short interfering RNA (siRNA) or single-stranded micro-RNA (miRNA) through the mechanisms of the RNA interference. (B) Initiating of protein synthesis by translating the information embedded in delivered mRNA into a protein with the participation of the intracellular machinery of protein synthesis.

**Figure 8 pharmaceutics-15-00194-f008:**
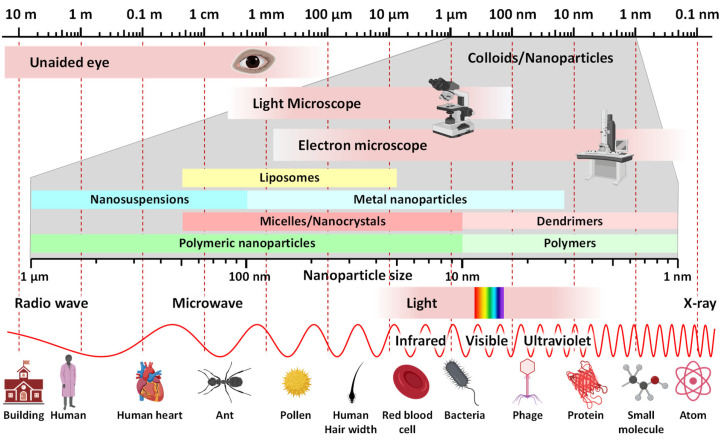
Nanoparticles/colloids are very small formations with a size between 1 nm (0.000001 mm) and 1 µm (0.001 mm). They cannot be seen with the naked eye nor can they be separated by the filtration through ordinary filter paper. Relatively large nanoparticles can be seen with a light microscope, while an electron microscope easily distinguishes nanoparticles of any dimensions. The size of nanoparticles is comparable to the wavelength of visible and ultraviolet light as well as the longest X-rays. They are significantly smaller than human cells and have a size similar to bacteria, phages, viruses and proteins. The size of nanoparticles that are used for the delivery of drugs and nucleic acids usually does not exceed 500 nm.

**Figure 9 pharmaceutics-15-00194-f009:**
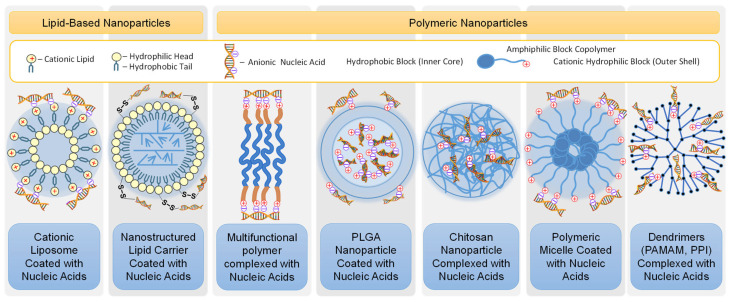
Main types of nanoparticles that can potentially be used for nucleic acid delivery.

**Figure 10 pharmaceutics-15-00194-f010:**
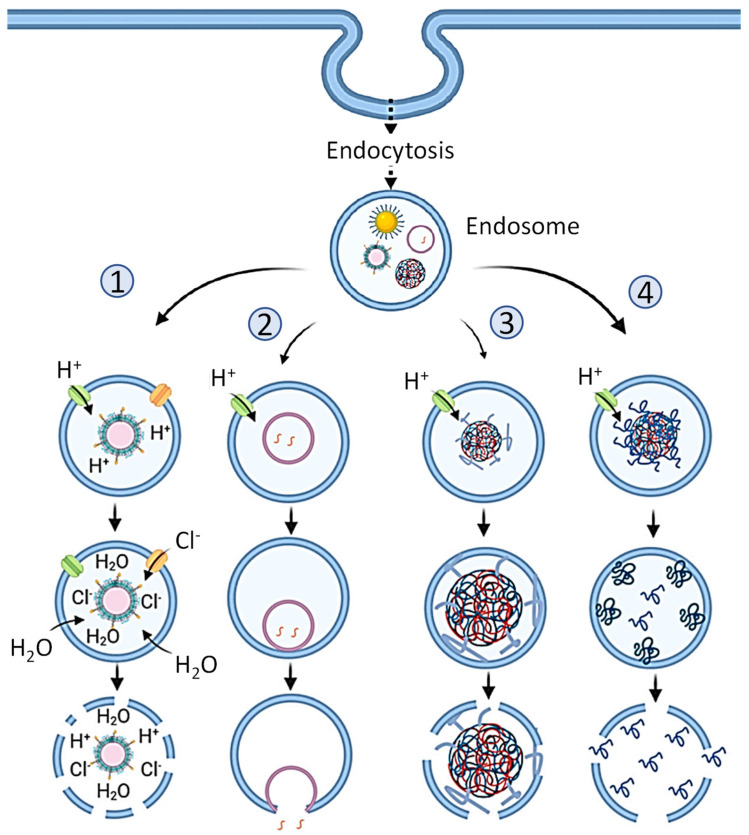
Four main types of endosomal escape mechanisms. After entering a cell by endocytosis, nanoparticles with encapsulated content are located inside endosomes—vesicle budded from the cellular plasma membrane. Four main possible mechanisms of disrupting the nanoparticles and escape of active components of the system from endosomes are usually hypothesized: (1) Proton-sponge effect and osmotic lysis [88]: When enclosed in acidic endosomes, weakly basic molecules work as proton sponges by absorbing protons in endosomes gradually increasing the membrane potential, forcing chloride to diffuse into the endosome and raising the osmotic pressure. Endosome swells and expands rupturing the lipid bilayer membrane and releasing the endosome contents into the cell. (2) Membrane fusion [89]: If nanoparticle membrane is formed by lipids (e.g., liposomes), it can merge through hydrophobic interactions with the endo/lysosomal membrane into a single continuous bilayer releasing the loaded cargo into the cytoplasm. (3) Particle swelling [90]: Maturation of endosomes and their fusion with lysosomes lowers the pH inside the organelle causing some nanoparticles (e.g., acidic responsive nanoparticles) to swell disrupting the vesicle membrane through increased mechanical strain. (4) Membrane translocation and destabilization [91]: The strong attraction of charged (cationic) compounds released from nanoparticles with negatively charged components of the inner membrane layer of endosomes as well as the release of pH-responsive polymers from nanoparticles may cause membrane destabilization, formation of transient pores and escape of nanoparticle materials and/or active content from endosomes Modified with permission from [87].

**Figure 11 pharmaceutics-15-00194-f011:**
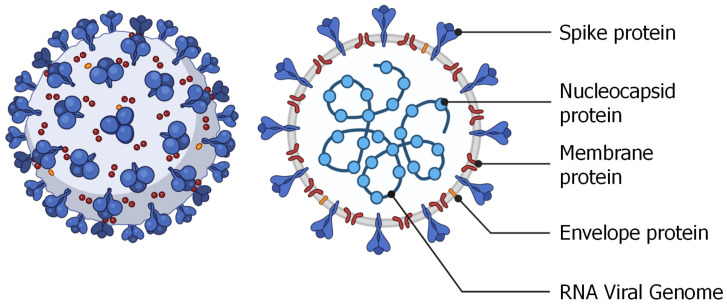
Schematic structure of SARS-CoV-2 virus. The genome of the virus represents a relatively large single-stranded positive-sense RNA formed the capsid with the nucleocapsid protein. This capsid is further packed by an envelope formed by three structural proteins: membrane protein, spike protein, and envelope protein. Redrawn from [10].

**Figure 12 pharmaceutics-15-00194-f012:**
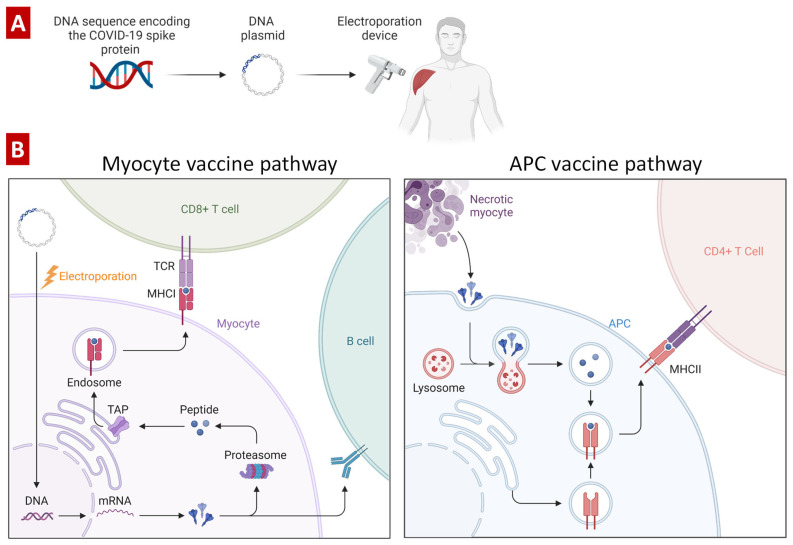
Two main types of DNA-based COVID-19 vaccines. (**A**) Intramuscular injection of pDNA encoding CoV spike protein using an electroporation device to promote its uptake by the cells near the injection. (**B**) Mechanisms of action of DNA-based COVID-19 vaccines. The peptides are transported to the endoplasmic reticulum with transport-associated proteins (TAP1 and TAP2) and subsequently bind to MHC class I molecules. After binding, MHC I is released from the complex and translocated via the Golgi apparatus to the cell surface, where it can be recognized by cytotoxic T cells (CD8+). Modified with permission from [106].

**Figure 13 pharmaceutics-15-00194-f013:**
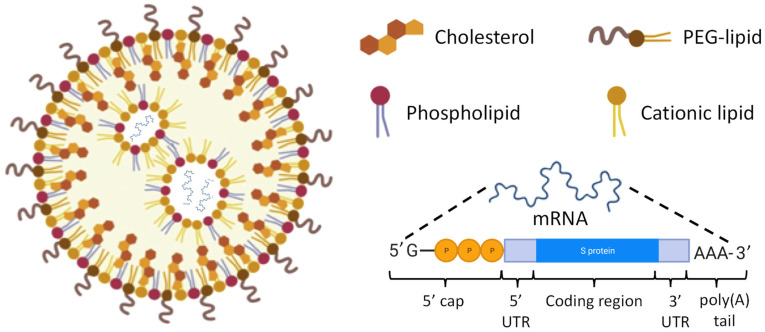
Typical structure of lipid nanoparticle-encapsulated mRNA encoding spike protein. Redrawn from [107].

**Figure 14 pharmaceutics-15-00194-f014:**
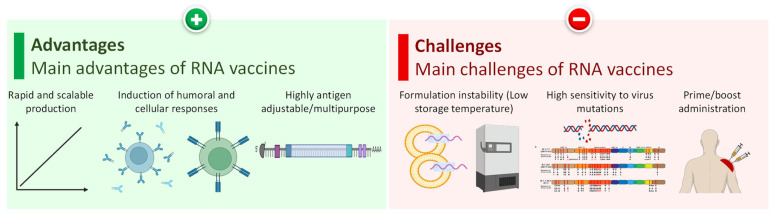
Advantages and challenges of RNA-based vaccines. Modified from [118].

## Data Availability

Not applicable.

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
