# Peer review of "Nanotechnology-Based RNA Vaccines: Fundamentals, Advantages and Challenges"

_pharmaceutics, 2023, doi:10.3390/pharmaceutics15010194_

Round 1

Reviewer 1 Report

The review " Nanotechnology-Based RNA Vaccines: Fundamentals, Ad-2 vantages and Challenges" deals with a hot topic of new class of RNA nano vaccines that made a revolution during the ongoing COVID-19 epidemic. It is thus very timely. It is written in an academic manner and covers most aspects of nanotechnology as related to these new vaccines. It is also beautifully illustrated. The authors underline that until recently, most of such vaccines were only at pre-clinical stages, due to the enormous complexity of the development and industrial production of such formulations leading to their high cost. Additionally, the development of nanomedicines requires joint efforts of multidisciplinary teams from such areas as pharmaceutical sciences, medicine, engineering, drug delivery, chemistry, molecular biology, physiology, etc. It also needs a deep understanding of a wide range of principles and mechanisms of immune response, nucleic acid functions, nanotechnology and vaccinations. These problems were successfully circumvented in a remarkably short time during the emergence of new coronavirus. The review provides a comprehensive account of the basic principles of work of the immune system, vaccination, nanotechnology and drug delivery in terms of the development and production of vaccines based on nanotechnology and the use of nucleic acids. An important part of the review is the analysis of reported problems and drawbacks associated with such vaccines and formulation of the ways to improve them, so that such rather rare side effects and problems should not be used by antivaxxers.

This reviewer has the following fairly minor concerns about the manuscript:

1. Lines 70-71. The authors may need to mention and reference inactivated vaccines, as well as those using viruses as delivery systems.

2. The sections 2.4.1 and 2.4.2 should be significantly reduced as they contain general information available elsewhere. This relates to Figs. 6 and 7 as well. At the same time, the lengthy description of the immune system may be left intact because of greater relevance to the review subject.

3. Section 2.5.1 might be trimmed down or removed.

4. The authors might like to add linear multifunctional polymers to the schematic on Fig. 10.

5. In section 4.3, please add a sentence about RNA stabilization in most advanced vaccines using the change of uridine to N1-methyl-pseudouridine (see for review Morais P, Adachi H, Yu YT. The critical contribution of pseudouridine to mRNA COVID-19 vaccines. Front Cell Dev Biol. 2021;9:789427. doi: 10.3389/fcell.2021.789427). Fig. 15 might be slightly modified to reflect this.

6. In Fig. 16, it is unclear why high sensitivity to virus mutations is a disadvantage. It pertains to all vaccines, yet, the RNA-based nanoformulations could be the easiest to change. The same concerns prime/boost, because this is the nature of immunity. At the same time, the authors might like to consider the resultant protein folding and thus, antigenicity problems that may require the presence of other viral proteins not represented in the single-gene vaccines.

7. The authors might like to cite a couple of recent papers that are in line with future directions:

a. Szebeni J, Storm G, Ljubimova JY, Castells M, Phillips EJ, Turjeman K, Barenholz Y, Crommelin DJA, Dobrovolskaia MA. Applying lessons learned from nanomedicines to understand rare hypersensitivity reactions to mRNA-based SARS-CoV-2 vaccines. Nat Nanotechnol. 202217:337-346. doi: 10.1038/s41565-022-01071-x.

b. Shi D, Beasock D, Fessler A, Szebeni J, Ljubimova JY, Afonin KA, Dobrovolskaia MA. To PEGylate or not to PEGylate: Immunological properties of nanomedicine's most popular component, polyethylene glycol and its alternatives. Adv Drug Deliv Rev. 2022;180:114079. doi: 10.1016/j.addr.2021.114079.

Author Response

Q: Lines 70-71. The authors may need to mention and reference inactivated vaccines, as well as those using viruses as delivery systems.

A: The references have been added.

Q: The sections 2.4.1 and 2.4.2 should be significantly reduced as they contain general information available elsewhere.

A: These sections have been reduced very significantly, leaving only information that vitally important for the delivery of nucleic acids and their use as vaccines.

Q: Section 2.5.1 might be trimmed down or removed.

A: Section 2.5.1 has been removed.

Q: The authors might like to add linear multifunctional polymers to the schematic on Fig. 10.

A: The linear multifunctional polymer has been added to the schematic on Fig. 10 (Figure 9 in the revised manuscript).

Q: In section 4.3, please add a sentence about RNA stabilization in most advanced vaccines using the change of uridine to N1-methyl-pseudouridine (see for review Morais P, Adachi H, Yu YT. The critical contribution of pseudouridine to mRNA COVID-19 vaccines. Front Cell Dev Biol. 2021;9:789427. doi: 10.3389/fcell.2021.789427).

A: Two sentences briefly describing N1-methyl-pseudouridine mRNA modification with the recommended citation have been added to the section (please see page 21).

Q: In Fig. 16, it is unclear why high sensitivity to virus mutations is a disadvantage. It pertains to all vaccines, yet, the RNA-based nanoformulations could be the easiest to change. The same concerns prime/boost, because this is the nature of immunity. At the same time, the authors might like to consider the resultant protein folding and thus, antigenicity problems that may require the presence of other viral proteins not represented in the single-gene vaccines.

A: We agreed with the Reviewer, that high sensitivity to virus mutations is pertains to all vaccines and that the RNA-based vaccines can quickly address virus mutations. We added recent data that confirm advantages of mRNA-based COVID-19 vaccines to address virus mutations. We also added a brief discussion of the influence of protein folding on antigenicity (Please see page 23).

The authors might like to cite a couple of recent papers that are in line with future directions:

  1. Szebeni J, Storm G, Ljubimova JY, Castells M, Phillips EJ, Turjeman K, Barenholz Y, Crommelin DJA, Dobrovolskaia MA. Applying lessons learned from nanomedicines to understand rare hypersensitivity reactions to mRNA-based SARS-CoV-2 vaccines. Nat Nanotechnol. 202217:337-346. doi: 10.1038/s41565-022-01071-x.
  2. Shi D, Beasock D, Fessler A, Szebeni J, Ljubimova JY, Afonin KA, Dobrovolskaia MA. To PEGylate or not to PEGylate: Immunological properties of nanomedicine's most popular component, polyethylene glycol and its alternatives. Adv Drug Deliv Rev. 2022;180:114079. doi: 10.1016/j.addr.2021.114079.

A: The mentioned references have been added to the manuscript.

Reviewer 2 Report

The review entitled “Nanotechnology-Based RNA Vaccines: Fundamentals, Advantages and Challenges” is a wide and comprehensive manuscript that tries to explain in a teaching way the basis of the immune system and how nanomedicine can modulate it. Indeed, the topics described are actually quite wide and therefore, in some cases the arguments are explained in a superficial way. The text is well written and organized. I appreciate the Authors’ attempt of providing a very clear and didactic description, however the Authors should consider that the audience of Pharmaceutics is already educated on these topics and the contribution of this review to the scientific world is not so relevant.

Specific comments:

1.      Please, check the affiliations, why did you use two numbers if the affiliation is the same for both Authors?

2.      Line 76, there are also scalability issue, please discuss also this aspect

3.      From line 82 to line 88. This part is redundant, it has already been said in section 1

4.      Line 149, check the font size all along the text

5.      Although I can understand the aim, the part about the immunity is too long. In particular, the explanation of the social distancing is completely superfluous.

6.      Also the section 2.4.1. is not necessary if we consider the audience of the journal. This section should be consistently reduced

7.      The section about nanomedicine is too much didactic, please avoid to provide naïve explanations  of what nanomedicine or medicine are

8.      Please explicit HPMI

The overall merit of this review is not really evident, it should have provided a more insight on this topic, therefore I suggest the Authors to make changes considering this aspect, otherwise I’ll be forced to reject the manuscript.

Author Response

Q: Please, check the affiliations, why did you use two numbers if the affiliation is the same for both Authors?

A: The numbers have been deleted from the affiliation.

Q: Line 76, there are also scalability issue, please discuss also this aspect.

A: Scalability issue discussion has been added (Please see page 3).

Q: From line 82 to line 88. This part is redundant, it has already been said in section 1

A: This part was removed.

Q: Line 149, check the font size all along the text

A: The font size was checked throughout the entire manuscript.

Q: Although I can understand the aim, the part about the immunity is too long. In particular, the explanation of the social distancing is completely superfluous.

A: This section has been significantly shortened. In particular, the explanation of social distancing and heard immunity (including the Figure 4) have been deleted from the manuscript.

Q: Also the section 2.4.1. is not necessary if we consider the audience of the journal. This section should be consistently reduced

A: Section 2.4.1. has been significantly reduced.

Q: The section about nanomedicine is too much didactic, please avoid to provide naïve explanations  of what nanomedicine or medicine are

A: Mentioned explanations have been deleted.

Q: Please explicit HPMI

A: We are sorry for the spelling error, HPMI polymer has been changed to N-(2-hydroxypropyl)methacrylamide (HPMA) copolymer.

Reviewer 3 Report

T. Minko and V.P. Pozharov gathered exciting information on RNA vaccines with a topic

entitled “Nanotechnology-Based RNA Vaccines: Fundamentals, Advantages,

and Challenges.” The authors provided basic surveillance mechanisms of the immune

system, nanotechnology in nucleic acid delivery, and their usage for the construction of

vaccines.

Overall, the review was well-conceived, well-written, and worth publishing in MDPI

Pharmaceutics. However, addressing the following major and minor issues is

recommended to reach more audiences and readers of different disciplines.

Major issues: Some of the descriptions are scientifically misleading.

1. The authors described small interfering (si)RNA. How can siRNA be applied to the

vaccine field?

To the best of my knowledge, no siRNA was utilized for the vaccine. For vaccination

effect, an antigen should be expressed [by introducing plasmid (p)DNA, messenger

(m)RNA], or an antigen [toxoid, antigen subunit] must be delivered. Instead, siRNA can

be used as an antiviral. This conflicts with the review title “Nanotechnology-Based RNA

Vaccines: Fundamentals, Advantages, and Challenges.”

2. The essence of the figure must be briefly written in the legends of the figure. So

that the readers will quickly grasp the scientific information.

Figure 11: Describe the gist of the figure in the legend.

Figure 13: Describe the gist of the figure in the legend.

Figure 14: Please see comment 10.

3. Line 73: The WHO compiles the COVID-19 vaccine landscape and its progress through

the pipeline. It is recommended to cite the following link, which will give every 3-month

update on vaccines.

https://www.who.int/publications/m/item/draft-landscape-of-covid-19-candidate-vaccines

4. Lines 98 and 99: The authors mentioned that “B-lymphocytes (B-cells), the immune

cells that produce antibodies, as well as natural killer (or NK) cells.”

Here, B-cells are not producing natural killer (or NK) cells. The information quoted

by the authors is scientifically misleading.

Natural Killer (NK) cells are lymphocytes in the same family as T- and B-cells and

originate from common progenitor cells, i.e., hematopoietic stem cells (HSCs) during

the transition from CD56high into CD56low; they undergo a progressive loss of NKG2A

and expression of KIRs, CD57, and NKG2C on terminally differentiated NK cells.

5. Lines 110 to 116: The authors described that RES consists of phagocytic cells. RES

also comprises the specialized endothelial cells of the liver, bone marrow, lymph

nodes, and spleen, which pinocytose the foreign materials and endogenous products

from the bloodstream. I recommend that the authors modify the description of RES

with suggested citations. The information shown below will be helpful to the authors.

RES is a network of immune cells comprising circulatory phagocytes (e.g., monocytes,

neutrophils, dendritic cells, etc.) and tissue-resident phagocytes (Kupfer cells in the

liver, alveolar macrophages in the lung, microglia in the brain, histiocytes in the

connective tissue, Red- and white-pulp macrophages, marginal zone macrophages,

and marginal zone metallophilic macrophages in the spleen, etc.,). In addition to these

immune cells, specialized endothelial cells (liver sinusoidal endothelial cells) play an

an essential role in the clearance of foreign particles/materials, viruses, and endogenous

soluble substances in the circulation and tissues (PLoS Pathog. 2011;7(9):e1002281, Sci Adv 2020;6(26):eabb8133, ACS Nano 2018 Mar 27;12(3):2088-2093.). The suggested articles demonstrated nature-derived viral nanoparticles (e.g., adenovirus) and artificially engineered nanoparticles (e.g., PEGylated DNA carriers and liposomes).

6. Line 816: What is the specific ligand in the context of the following sentence described

by the authors?

“Biopolymers such as HPMI polymer and use specific ligands specific to dendritic

cells.”

Change the specific ligands à mannose

7. Figure 5: Viral vectors also belong to the family of nucleic acid carriers. Hence, it is

recommended to separate them with proper naming.

Naked nucleic acid vaccine, nonviral nucleic acid vaccine (mRNA-loaded lipid

nanoparticle, mRNA complexed liposome, etc.)

8. Figure 10: Better to change the name of polymeric micelle to cationic polymeric micelle

surface coated with nucleic acids.

9. Figure 10: Better to change the name of the cationic liposome to cationic liposome

coated with nucleic acids. Cationic liposomes were used to complex the nucleic acids,

namely RNA-Lipoplexes. Here, we can tune the surface charge of the RNA-lipoplex to

cationic or anionic even after coating the nucleic acid by adjusting the ratio of positive

to negative charges.

10. Figure 11: Endosomal trapping is one of the significant bottlenecks toward

maximizing the cytosolic delivery of nucleic acids and drugs that show activity in the

cellular compartments such as the nucleus and mitochondria. It would be interesting

to give at least one reference for each type of endosomal escape and a brief

discussion in the figure legends.

For example:

1. Proton-sponge and osmotic lysis à (Proc Natl Acad Sci U S A 1995;92(16):7297-

301). The acidified endosomal and lysosomal compartments increase the influx of

chloride counterions and lyse the membrane of endosomes and lysosomes due to

increased osmotic pressure.

2. Membrane fusion à (Biomaterials 2009;30(15):2940-9). Membrane fusion

between the nanoparticle and endo/lysosomal membrane releases the loaded

cargo into the cytoplasm.

3. Particle swelling à (Nano Lett 2007;7(10):3056-64). The swelling of acidic responsive

nanoparticles rupture the membrane of endosomes through

increased mechanical strain.

4. Membrane translocation and destabilization à (Macromol Rapid Commun

2022;43(12):e2100754). The release of pH-responsive polymers from nanoparticles facilitates the endo/lysosomal escape by destabilizing the membrane of endo/lysosomal compartments.

11. Figure 14: What are TAP and TCR?

Expand the acronyms and provide the gist of the figure as the legend.

For example:

The peptides are transported to the endoplasmic reticulum with transport-associated

proteins (TAP1 and TAP2) and subsequently bind to MHC class I molecules. After

binding, MHC I is released from the complex and translocated via the Golgi

apparatus to the cell surface, where it can be recognized by cytotoxic T cells (CD8+).

12. Figure 14: The authors showed the figure as follows.

Change the CoV spike protein à DNA sequence encoding CoV spike protein

Here, the meaning is conveyed as CoV spike protein is inserted into plasmid DNA.

In fact, the DNA sequence encoding the CoV spike protein is inserted.

13. Figure 14: Separate A (upper part) and B (lower part) of Figure 14.

14. Figure 14 A: Intramuscular injection of pDNA encoding CoV spike protein using an

electroporation device to promote its uptake by the cells near the injection.

15. Figure 14 B: The gist of the figure is necessary.

16. Line 719: The author described, "Being internalized by myocytes, plasmid stimulates

the synthesis of the spike protein.” Here, the meaning is that muscle cells are

transfected. Indeed, besides myocytes, tissue-resident macrophages or recruited

antigen-presenting cells (dendritic cells) were also transfected. Please modify the

description accordingly (Adv Drug Deliv Rev 2021;170:83-112 and Adv Drug Deliv Rev 2020;158:91-115).

17. The use of delivery systems for nucleic acid vaccines needs to be more convincing.

The authors emphasized siRNA. Lines 648, 649: Previously, we showed that naked

non-conjugated RNA degrades in human serum within minutes (Figure 12) [53,89].

However, the degradation kinetics of mRNA and pDNA differ from the siRNA. It

may be better to emphasize that enzymatic degradation is one of the issues for

efficient nucleic acid delivery. Later, emphasize the importance of delivery

systems/nanoparticles for each nucleic acid with proper citation.

Loading the mRNA into a polyplex micelle protected the mRNA from enzymatic

degradation by 10,000-fold compared to naked mRNA (J Drug Target 2019;27(5-

6):670-680, Mol Pharm 2018 Jun 4;15(6):2268-2276.).

The naked pDNA was rapidly digested in the serum (Pharm Res 1995;12(6):825-30.), whereas complexing it within the polyplex micelle substantially protected it from enzymatic degradation. (Biomaterials 2014;35(20):5359-5368).

18. Figure 12: Figure 12 demonstrates the importance of nanoparticles for protecting the

siRNA.

How is siRNA used in vaccines? I think siRNA can be used as an antiviral. In the

following section, the authors described nucleic acid vaccines “3. Nanotechnology-

Based Nucleic Acid Vaccines”. The authors must connect the review story with

mRNA and pDNA, rather than siRNA.

19. Future directions:

The authors only focused on only formulation and stability issues of vaccines. In

addition to these issues, one of the significant issues of lipid nanoparticle-loaded

mRNA vaccines is off-targeting to the liver even though the LNP-mRNA vaccine is

injected via intramuscular injection (Cell 2020;182(5):1271-1283.e16 and

 J Control Release 2015;217:345-51). Such liver off-targeting may cause

toxicity/immunogenicity and inflammation exacerbation based on the type of antigen that is expressed (J Control Release 2022;344:50-61). Minimizing hepatic off-targeting while maximizing the delivery at the injection site might be advantageous for widespread vaccine applications (Curr Opin Virol 2021;48:65-72). Discussion in this direction might be interesting.

Minor issues:

1. Redundant acronyms and abbreviations were observed throughout the manuscript. It

may distract the focus of readers. It is generally recommended to provide an acronym

and abbreviation after their first use in the manuscript.

Lines 99, 159, and 180: natural killer (NK) cells

Lines 205, 221: antigen-presenting cells (APC)

2. It may not be required to have an acronym and abbreviation if the description is

provided only once in the manuscript.

For example:

Line 222: dendritic cells (DCs)

Line 601: poly(lactic-co-glycolic) acid (PLGA)

Line 749: self-amplifying RNA (saRNA)

3. Line 265: vs à versus

4. Spelling mistakes were observed throughout the text.

Line 589: careers à carriers

Line 741: COVID-19 spice protein —> COVID-19 spike protein.

Line 601: poly(lactic-co-glycolic) acid à poly(lactic-co-glycolic) acid

Line 691: In this section —> In the following section

5. Expand the complete form of an acronym or abbreviation.

For example,

Line 703: ACE2 à angiotensin-converting enzyme-2 (ACE-2)

Line 816: HPMI à N-(4-Hydroxy phenyl) maleimide [Here, the locant should be

italic].

6. Gaps were observed.

Line 768: Disadvan tages à Disadvantages

7. A few letters were italicized.

For example:

Line 796: viable à viable

8. Use consistent terminology:

Sometimes spike protein, sometimes S protein was used. Please unify the spike

protein (S protein) after its first appearance in the text.

Author Response

Q: The authors described small interfering (si)RNA. How can siRNA be applied to the vaccine field?To the best of my knowledge, no siRNA was utilized for the vaccine. For vaccination effect, an antigen should be expressed [by introducing plasmid (p)DNA, messenger (m)RNA], or an antigen [toxoid, antigen subunit] must be delivered. Instead, siRNA can be used as an antiviral. This conflicts with the review title “Nanotechnology-Based RNA Vaccines: Fundamentals, Advantages, and Challenges.”

A: Initially, we present data for siRNA encapsulation into nanoparticles to show how nanotechnology can prevent the degradation of nucleic acids in body fluids. According to the Reviewer’s recommendation, the figure (Figure 11) has been deleted from the review. However, we left the comparison of intracellular action siRNA/miRNA and RNA in order to support the use of only a single-stranded coding RNA as a vaccine and underline its mechanisms of intracellular action.

Q: The essence of the figure must be briefly written in the legends of the figure. So that the readers will quickly grasp the scientific information.

Figure 11: Describe the gist of the figure in the legend.

A: A more detailed description has been added to the Figure 11 (Figure 10 in the revised manuscript).

Figure 13: Describe the gist of the figure in the legend.

A: A description has been added to the Figure 13 (Figure 11 in the revised manuscript).

Q: Line 73: The WHO compiles the COVID-19 vaccine landscape and its progress through the pipeline. It is recommended to cite the following link, which will give every 3-month update on vaccines.

https://www.who.int/publications/m/item/draft-landscape-of-covid-19-candidate-vaccines

A: The recommended citation has been added (Please see page 2, reference 21).

Q: Lines 98 and 99: The authors mentioned that “B-lymphocytes (B-cells), the immune cells that produce antibodies, as well as natural killer (or NK) cells.” Here, B-cells are not producing natural killer (or NK) cells. The information quoted by the authors is scientifically misleading. Natural Killer (NK) cells are lymphocytes in the same family as T- and B-cells and originate from common progenitor cells, i.e., hematopoietic stem cells (HSCs) during the transition from CD56high into CD56low; they undergo a progressive loss of  NKG2A and expression of KIRs, CD57, and NKG2C on terminally differentiated NK cells.

A: The suggested explanation has been added to the text of the revised manuscript (please see page 3).

Q: Lines 110 to 116: The authors described that RES consists of phagocytic cells. RES also comprises the specialized endothelial cells of the liver, bone marrow, lymph nodes, and spleen, which pinocytose the foreign materials and endogenous products from the bloodstream. I recommend that the authors modify the description of RES with suggested citations. The information shown below will be helpful to the authors. RES is a network of immune cells comprising circulatory phagocytes (e.g., monocytes, neutrophils, dendritic cells, etc.) and tissue-resident phagocytes (Kupfer cells in the liver, alveolar macrophages in the lung, microglia in the brain, histiocytes in the connective tissue, Red- and white-pulp macrophages, marginal zone macrophages, and marginal zone metallophilic macrophages in the spleen, etc.,). In addition to these immune cells, specialized endothelial cells (liver sinusoidal endothelial cells) play an an essential role in the clearance of foreign particles/materials, viruses, and endogenous soluble substances in the circulation and tissues (PLoS Pathog. 2011;7(9):e1002281, Sci Adv 2020;6(26):eabb8133, ACS Nano 2018 Mar 27;12(3):2088-2093.). The suggested articles demonstrated nature-derived viral nanoparticles (e.g., adenovirus) and artificially engineered nanoparticles (e.g., PEGylated DNA carriers and liposomes).

A: The recommended modification has been made and the suggested citations have been added (please see pages 3-4).

Q: Line 816: What is the specific ligand in the context of the following sentence described by the authors?

“Biopolymers such as HPMI polymer and use specific ligands specific to dendritic cells.”

Change the specific ligands à mannose

A: The suggested change has been made (please see page 23).

Q: Figure 5: Viral vectors also belong to the family of nucleic acid carriers. Hence, it is recommended to separate them with proper naming. Naked nucleic acid vaccine, nonviral nucleic acid vaccine (mRNA-loaded lipid nanoparticle, mRNA complexed liposome, etc.) 

A: The suggested changes have been made in the former Figure 5 (Figure 4 in the revised manuscript).

Q: Figure 10: Better to change the name of polymeric micelle to cationic polymeric micelle surface coated with nucleic acids.

A: The suggested change has been made in the former Figure 10 (Figure 9 in the revised manuscript).

Q: Figure 10: Better to change the name of the cationic liposome to cationic liposome coated with nucleic acids. Cationic liposomes were used to complex the nucleic acids, namely RNA-Lipoplexes. Here, we can tune the surface charge of the RNA-lipoplex to cationic or anionic even after coating the nucleic acid by adjusting the ratio of positive to negative charges.

A: The suggested change has been made in the former Figure 10 (Figure 9 in the revised manuscript).

Q: Figure 11: Endosomal trapping is one of the significant bottlenecks toward maximizing the cytosolic delivery of nucleic acids and drugs that show activity in the cellular compartments such as the nucleus and mitochondria. It would be interesting to give at least one reference for each type of endosomal escape and a brief discussion in the figure legends.

For example:

  1. Proton-sponge and osmotic lysis à (Proc Natl Acad Sci U S A 1995;92(16):7297- 301).
  2. The acidified endosomal and lysosomal compartments increase the influx of chloride counterions and lyse the membrane of endosomes and lysosomes due to increased osmotic pressure. Membrane fusion à (Biomaterials 2009;30(15):2940-9). Membrane fusion between the nanoparticle and endo/lysosomal membrane releases the loaded cargo into the cytoplasm.
  1. Particle swelling à (Nano Lett 2007;7(10):3056-64). The swelling of acidic responsive nanoparticles rupture the membrane of endosomes through increased mechanical strain.
  1. Membrane translocation and destabilization à (Macromol Rapid Commun 2022;43(12):e2100754). The release of pH-responsive polymers from nanoparticles facilitates the endo/lysosomal escape by destabilizing the membrane of endo/lysosomal compartments.

A: The suggested explanations and citations have been added to the Figure 11 (Figure 10 in the revised manuscript). 

Q: Figure 14: What are TAP and TCR? Expand the acronyms and provide the gist of the figure as the legend.

For example: 

The peptides are transported to the endoplasmic reticulum with transport-associated proteins (TAP1 and TAP2) and subsequently bind to MHC class I molecules. After binding, MHC I is released from the complex and translocated via the Golgi apparatus to the cell surface, where it can be recognized by  cytotoxic T cells (CD8+).

A: The proposed explanations have been added to the Figure 14 (Figure 12 in the revised manuscript). 

Q: Figure 14: The authors showed the figure as follows. Change the CoV spike protein à DNA sequence encoding CoV spike protein Here, the meaning is conveyed as CoV spike protein is inserted into plasmid DNA. In fact, the DNA sequence encoding the CoV spike protein is inserted. 

A: The proposed explanations have been added to the Figure 14 (Figure 12 in the revised manuscript). 

Q: Figure 14: Separate A (upper part) and B (lower part) of Figure 14.

Figure 14 A: Intramuscular injection of pDNA encoding CoV spike protein using an electroporation device to promote its uptake by the cells near the injection.

Figure 14 B: The gist of the figure is necessary.

A: The proposed changes have been made in the Figure 14 (Figure 12 in the revised manuscript). 

Q: Line 719: The author described, "Being internalized by myocytes, plasmid stimulates the synthesis of the spike protein.” Here, the meaning is that muscle cells are transfected. Indeed, besides myocytes, tissue-resident macrophages or recruited antigen-presenting cells (dendritic cells) were also transfected. Please modify the description accordingly (Adv Drug Deliv Rev 2021;170:83-112 and Adv Drug Deliv Rev 2020;158:91-115).

A: The description has been modified as advised and proposed citations have been added (please see page 20).

Q: The use of delivery systems for nucleic acid vaccines needs to be more convincing. The authors emphasized siRNA. Lines 648, 649: Previously, we showed that naked non-conjugated RNA degrades in human serum within minutes (Figure 12) [53,89]. However, the degradation kinetics of mRNA and pDNA differ from the siRNA. It may be better to emphasize that enzymatic degradation is one of the issues for efficient nucleic acid delivery. Later, emphasize the importance of delivery systems/nanoparticles for each nucleic acid with proper citation. Loading the mRNA into a polyplex micelle protected the mRNA from enzymatic degradation by 10,000-fold compared to naked mRNA (J Drug Target 2019;27(5-6):670-680, Mol Pharm 2018 Jun 4;15(6):2268-2276.). The naked pDNA was rapidly digested in the serum (Pharm Res 1995;12(6):825-30.), whereas complexing it within the polyplex micelle substantially protected it from enzymatic degradation. (Biomaterials 2014;35(20):5359-5368).

A: The proposed changes have been added with corresponding citations to the text of the revised manuscript (Please see page 18).

Q: Figure 12: Figure 12 demonstrates the importance of nanoparticles for protecting the siRNA. How is siRNA used in vaccines? I think siRNA can be used as an antiviral. In the following section, the authors described nucleic acid vaccines “3. Nanotechnology-Based Nucleic Acid Vaccines”. The authors must connect the review story with mRNA and pDNA, rather than siRNA.

A: The Figure 12 and discussion of siRNA stability has been removed from the manuscript.

Q: Future directions: The authors only focused on only formulation and stability issues of vaccines. In addition to these issues, one of the significant issues of lipid nanoparticle-loaded mRNA vaccines is off-targeting to the liver even though the LNP-mRNA vaccine is injected via intramuscular injection (Cell 2020;182(5):1271-1283.e16 and  J Control Release 2015;217:345-51). Such liver off-targeting may cause toxicity/immunogenicity and inflammation exacerbation based on the type of antigen that is expressed (J Control Release 2022;344:50-61). Minimizing hepatic off-targeting while maximizing the delivery at the injection site might be advantageous for widespread vaccine applications (Curr Opin Virol 2021;48:65-72). Discussion in this direction might be interesting.

A: The recommended discussion with proper citations has been added (Please see page 23).

 Minor issues:

Q: Redundant acronyms and abbreviations were observed throughout the manuscript. It may distract the focus of readers. It is generally recommended to provide an acronym and abbreviation after their first use in the manuscript.

Lines 99, 159, and 180: natural killer (NK) cells

Lines 205, 221: antigen-presenting cells (APC)

A: Redundant acronyms and abbreviations have been deleted from the text of the manuscript.

Q: It may not be required to have an acronym and abbreviation if the  description is provided only once in the manuscript.

For example:

Line 222: dendritic cells (DCs)

Line 601: poly(lactic-co-glycolic) acid (PLGA)

Line 749: self-amplifying RNA (saRNA)

A: The mentioned acronyms have been removed from the text of the  manuscript.

Q: Line 265: vs à versus

A: Vs has been changed to versus (please see page 7).

Q: Spelling mistakes were observed throughout the text.

Line 589: careers à carriers

Line 741: COVID-19 spice protein —> COVID-19 spike protein.

Line 601: poly(lactic-co-glycolic) acid à poly(lactic-co-glycolic) acid

Line 691: In this section —> In the following section

A: The mentioned spelling mistakes have been corrected.

Q: Expand the complete form of an acronym or abbreviation.

For example,

Line 703: ACE2 à angiotensin-converting enzyme-2 (ACE-2) 

Line 816: HPMI à N-(4-Hydroxy phenyl) maleimide [Here, the locant should be

italic].

A: The complete forms of acronyms have been expanded.

Q: Gaps were observed.

Line 768: Disadvan tages à Disadvantages

A: The gaps have been removed.

Q: A few letters were italicized.

For example:

Line 796: viable à viable

A: Unnecessary italicized letters have been changed to the regular font.

Q: Use consistent terminology:

Sometimes spike protein, sometimes S protein was used. Please unify the spike protein (S protein) after its first appearance in the text.

A: The abbreviation “S protein” has been removed.

Round 2

Reviewer 2 Report

The Auhtors have made all the required corrections and no more comments are needed.

Reviewer 3 Report

 Accepted in present form